# Relationships between Anxiety, Repetitive Behavior and Parenting Stress: A Comparative Study between Individuals with Autism from Spain and Colombia

**DOI:** 10.3390/brainsci14090910

**Published:** 2024-09-09

**Authors:** Tíscar Rodríguez-Jiménez, Agustín E. Martínez-González

**Affiliations:** 1Area of Personality, Assessment and Psychological Treatments, Department of Psychology and Sociology, Faculty of Social and Human Sciences, University of Zaragoza, 50009 Teruel, Spain; trodriguez@unizar.es; 2Department of Developmental Psychology and Didactics, University of Alicante, 03690 Alicante, Spain

**Keywords:** autism spectrum disorder, anxiety, repetitive behavior, parental stress

## Abstract

The present study analyzed the association between anxiety, repetitive behavior and parental stress in individuals with autism from Spain (*n* = 60, mean age = 8.52, SD = 4.41) and Colombia (*n* = 58, mean age = 10.29, SD = 4.98). Similarly, differences in anxiety, repetitive behavior and parental stress between both countries were analyzed. Outcomes revealed a strong relationship between anxiety and repetitive behavior in both populations. Furthermore, moderate positive associations were observed between anxiety, repetitive behavior and parental stress in the Spanish sample. However, parental stress was found to be moderately and negatively related with anxiety and repetitive behavior in the Colombian sample. Finally, no differences were found in anxiety and repetitive behavior between countries, but differences did emerge for parental stress which was found to be higher in the Colombian sample. In conclusion, differences in parental stress may be due to regional differences in socio-health resources, socio-economics, parenting styles, etc.

## 1. Introduction

Autism spectrum disorder (ASD) is a neurodevelopmental disorder characterized by difficulties with social communication, poor interaction skills and the presence of restricted and repetitive behavioral patterns [1]. The following different types of repetitive behaviors (RRBs) exist: stereotyped, self-injurious, compulsive, ritualistic and repetitive actions [2].

The prevalence of ASD in both women and men has increased around the world, with a 312% rise in the US since 2000. In other words, one in every 36 inhabitants in the US is diagnosed with ASD, with similar trends also being detected in other developed countries [3,4,5,6]. Notably, there is a lower incidence in developing countries such as Asia (0.36%) [6] and India (0.11%) [7], or in Latin countries such as Venezuela (0.17%) [8], Brazil (0.27%) [9] and Mexico (0.87%) [10]. An important knowledge gap is observed that may be due to the lack of standardized evaluation measures and the difficulty of performing evaluations in rural environments [7]. In fact, more developed Latin American countries, such as Chile, present prevalence rates that are similar to other European countries, such as Spain (1.95%) [11], while in Colombia, no rigorous studies are available due to the lack of prior validation studies using relevant instruments [12,13]. Thus, the rural and urban context, as well as the educational context and the country’s resources, etc., must be considered as important study variables when assessing autism severity [14]. To this end, there has been an exponential increase in publications on autism in different fields of study [15,16,17], as well as a surge in research on the validation of relevant instruments that serve the aim of achieving an average global diagnosis age of 5 years [18].

A meta-analysis including 4459 young people with autism reported that 33% reported having anxiety symptoms, whilst 19% reported having an anxiety disorder [19]. These figures point to the importance of emotional regulation in ASD. Another meta-analysis that included measures of anxiety and the intelligence quotient within 18,430 children with ASD found that children with a higher intelligence quotient also scored higher on anxiety measures. However, many studies used anxiety measures that had not been previously validated in the country of administration, especially in those relevant to the ASD population with intellectual disabilities (IDs) [20].

Several psychobiological factors may be associated with anxiety in autism. The first possible factor refers to the intolerance of uncertainty, in that it has been shown that anxiety and intolerance of uncertainty are consistently elevated in individuals with ASD [21]. Another factor closely associated with anxiety in autism is hypersensitivity to certain stimuli [22,23,24,25,26]. This hypersensitivity can subsequently manifest itself as an increase in the severity of RRBs [1,27,28]. Previous studies have reported small to moderate correlations between RRBs and anxiety [29]. Specifically, insistence on sameness correlates more strongly with anxiety in individuals with ASD [22,30,31,32]. Similarly, a clear association has been found between emotional instability and self-injury. It seems that stereotyped behavior is more strongly associated with moderate anxiety and social communication [33]. All of the aforementioned findings are consistent with the “continuous emotional state-repetitive behavior in ASD” theoretical model conceived by Martínez-González et al. [33]. This model suggests that the repetitiveness of behaviors reflecting similarity and stereotypes is associated with lower anxiety levels and increasing adaptation to the environment [34], whilst self-injury is associated with high levels of anxiety, emotional lack of control and social adaptation problems [35,36]. Finally, sensory hyper-reactivity can cause selective or restrictive eating patterns (e.g., being a very picky eater) [37,38], which, in turn, can generate an alteration of the gut-microbiota and lead to the emergence of gastro-intestinal symptoms (e.g., abdominal pain, constipation, etc.) in ASD samples [15,37,39]. Furthermore, gastro-intestinal symptoms lead to high levels of parental stress [40].

Similarly, a relationship has been found to exist between externalizing symptoms, including RRBs, internalizing symptoms, such as anxiety in children with ASD, and parental stress in caregivers [41,42,43,44]. Thus, the emotional instability of individuals with ASD generates parental stress and causes a decrease in the family’s quality of life [45]. The emotional development of children is a dynamic process over the years and the parent–child relationship is a crucial factor in their mental health [46]. Studies suggest that high levels of anxiety in individuals with ASD are associated with greater parental stress, which manifests as anxiety and depression in families [47,48,49]. Thus, some studies have found high rates of anxiety and depression, specifically, between 40.1% and 51%, in parents of children with ASD [40,50]. A meta-analysis found four factors to be associated with parental stress. Namely, these factors were parental sex, diagnosis-related coping issues, socioeconomic characteristics and parental social isolation. This study concluded by arguing that it is necessary to implement ongoing social policies and interventions as a means of providing diagnosis-related support [51]. Additionally, there appears to be a significant amount of family distress that affects both parents and siblings. Depressive symptoms are most frequent in mothers of children with ASD [52].

Although an increase has been seen in the number of publications on autism, few cross-cultural studies have been conducted [53]. When perusing previous literature, more cross-cultural studies have been conducted on parental stress than on anxiety in individuals with ASD [54,55]. Cross-cultural studies on parental stress in autistic samples have produced mixed results. On the one hand, Japanese parents exhibited higher parental stress and a less engaging social style than did Italian parents. Additionally, Japanese parents with a child with more severe ASD were more likely to experience higher levels of parental stress [54]. Other studies have found similar patterns when comparing European and Asian countries, with parental stress being higher in Chinese caregivers of autistic children than in Dutch caregivers [56]. On the other hand, no significant differences were found between US and non-Western countries (e.g., India and Japan) in terms of maternal parental stress [55,57]. A cross-cultural study between India and the US found that parental stress acted as a predictor of children’s communication skills, with this relationship being moderated by parent’s cultural orientation. Specifically, findings revealed a negative correlation between children’s communication skills and parental stress [57]. However, there is a paucity of comparative studies on anxiety in children with ASD and parental stress among Spanish-speaking countries. A comparison of two Spanish-speaking countries with similar cultures, namely, Spain and Colombia, indicated that the Colombian ASD population presents higher levels of RRBs [12]. Cultural and epigenetic factors may influence the severity of ASD [15,58,59].

In conclusion, only a limited number of cross-cultural studies exist on parental anxiety and stress in autistic samples. Most studies have compared Western and Eastern cultures, with very few focusing on Latin American populations [54,55]. Typically, studies include samples of between 20 and 100 caregivers who have a child with ASD. It is therefore necessary to examine parental anxiety and stress profiles in ASD samples in Spanish-speaking countries. Likewise, in the analysis of the relationships between hyperreactivity, RRBs and anxiety, previous researchers have had a limited sample which could only be between 40 and 70 families [22,24,25,26]. Similarly, studies with similar sample sizes have been published for the analysis of relationships between anxiety, RRBs and parental stress in autism [43,44].

The present study examined the relationships between anxiety, repetitive behavior and parental stress in individuals with ASD from Spain and Colombia, whilst also examining differences between both countries.

## 2. Materials and Methods

### 2.1. Participants

A sample of 118 families comprising individuals with ASD (60 Spanish and 58 Colombian) participated in the study survey. Participants came from all regions of Spain and Colombia. Their children were enrolled in educational centers (e.g., regular education, special education, etc.) or daycare centers. The Spanish participants belonged to the following regions: Murcia, Valencia and Andalusia. The Colombian sample came from the Andean Caribbean, Orinoquía and Pacific regions.

DSM-5 diagnostic criteria for ASD [60] formed the basis of the inclusion criteria applied to both Spanish and Colombian samples. Spanish and Colombian participants had previously been diagnosed by mental health institutions (e.g., diagnostic services of each country).

Individuals with other concomitant diagnoses (e.g., motor disabilities, multiple disabilities, attention-deficit/hyperactivity disorder, etc.) were excluded. Participants with ASD and intellectual disabilities (IDs) were included when ASD was the primary diagnosis. Table 1 shows the sociodemographic and diagnostic characteristics of the sample.

### 2.2. Measurements

#### 2.2.1. Sociodemographic Questionnaire

Lam and Aman’s sociodemographic questionnaire [61] was used for the pre-sent study. Information was collected on (a) sociodemographics (e.g., sex, age and country), (b) the main diagnosis according to DSM criteria and (c) comorbidities with other neurodevelopmental disorders.

#### 2.2.2. Social Communication Questionnaire, SCQ Form B (SCQ-B)

The SCQ-Form B [62,63] is a scale targeted towards parents or caregivers. It comprises 40 items that are designed to determine the potential presence of ASD. It produces an overall score and a maximum of three possible additional scores (social interaction problems, communication difficulties and restricted, repetitive and stereotyped behaviors).

Administration typically takes around ten minutes. The SCQ-B assesses autism symptoms during the past three months. The cut-off point is scores above 15. Overall SCQ-B scores were used in the present study. This scale has shown adequate psychometric properties [63]. Internal consistency of the overall score was 0.76 in the Spanish sample and 0.70 in the Colombian sample.

#### 2.2.3. Repetitive Behavior Scale-Revised (RBS-R)

The RBS-R evaluates repetitive behaviors in individuals with ASD and IDs through 43 items grouped into 6 different dimensions (stereotypic, self-injurious, compulsive, ritualistic, sameness and restrictive behaviors) [2]. Responses are recorded along a 4-point rating scale ranging from 0 (repetitive behavior does not occur) to 3 (very severe repetitive behaviors). Assessment of repetitive behavior is based on observations of and interactions with the respondent during the month prior to completion. The RBS-R has demonstrated adequate psychometric properties for use with individuals with ASD from different countries [64,65,66,67,68,69,70,71,72,73]. Internal consistency of RBS-R in the autistic Spanish sample (α = 0.95) was high for the majority of subscales: stereotyped (α = 0.81), self-injurious (α = 0.77), compulsive (α = 0.86), ritualistic (α = 0.85), sameness (α = 0.86) and restricted behaviors (α = 0.74) [71]. Internal consistency of the RBS-R in the Colombian sample was higher: stereotyped (α = 0.83), self-injurious (α = 0.87), compulsive (α = 0.92), ritualistic (α = 0.89), sameness (α = 0.94), restricted behaviors (α = 0.78) and overall RBS-R (α = 0.97) [12].

#### 2.2.4. Parent-Rated Anxiety Scale (PRAS-ASD)

The PRAS-ASD represents a reliable and valid scale for measuring anxiety in youth with ASD. It consists of 25 questions related to anxiety, with question responses ranging from 0 (none) to 3 (severe). The scale describes a single factor and is administered to caregivers. An α coefficient of 0.93 has previously been reported [31]. In the present study, internal consistency was 0.94 and 0.96 in the Spanish and Colombian samples, respectively.

#### 2.2.5. Parenting Stress Index-Short Form (PSI-SF)

This scale is a reduced version of the parenting stress index (PSI) [74], which consists of 36 items, divided into three subscales: parental distress (PD), difficult child (DC) and parent–child dysfunctional interaction (P-CDI). Overall parenting stress (GPS) scores can also be calculated. The form uses a 5-point Likert scale (1 = strongly disagree; 5 = strongly agree), with higher scores indicating higher parenting stress. Cronbach alpha outcomes for this scale range from 0.55 to 0.80 [74]. Furthermore, the PSI-SF has been used with both ASD and neurotypical populations [75]. In the present study, internal consistency values for the Spanish sample were 0.84 for parental distress, 0.77 for parent–child dysfunctional interaction, 0.85 for difficult child and 0.91 for global parenting stress. On the other hand, internal consistency values for the Colombian sample were 0.91 for parental distress, 0.91 for parent–child dysfunctional interaction, 0.80 for difficult child and 0.95 for global parenting stress.

### 2.3. Procedures and Ethics

Initial assessment and diagnosis: All participating families and caregivers had a child diagnosed with ASD according to DSM-5 criteria [60]. Individuals with ASD with or without IDs were diagnosed according to DSM-5 criteria using standardized scales (e.g., the Wechsler Nonverbal Scale of Ability and Leiter-3 scale). All participants had been previously diagnosed by the pertinent mental health services and institutions in each country responsible for establishing the degree of disability and dependency. The initial assessment and diagnosis of autism was confirmed by accredited experts from the health system of each country. With regards to Spanish participants, all diagnoses had previously been made at early care centers and pediatric services run by regional mental health services [76]. Common standardized scales were used to diagnose ASD. However, the diagnostic process of individuals with ASD in Colombia adhered to the “clinical protocol for the diagnosis, treatment and comprehensive care route for boys and girls with autism spectrum disorders” outlined by the Colombian Ministry of Health [77]. This national protocol employs applied behavior analysis (ABA) to determine a clinical and non-psychometric diagnosis. Diagnoses in Colombia were limited by the fact that most standardized scales have not yet been validated and most commonly used scales come from a Spanish or American context. Furthermore, not all participating centers had psychometric scales available.

Survey administration: In Spain and Colombia, a cover letter containing study information was sent to participating ordinary centers, special education centers, associations of families with children with autism, etc. Researchers organized a training session for all the participating centers, at which they described the purpose of the research study, tests to be used and provided instructions for test administration. Many of the tests included in the protocol were known by the professionals at the participating centers. A two-hour session was held to explain the entire protocol and resolve procedural questions to the centers’ professionals. Participating centers made appointments with potential participants via an online meeting and individual phone calls in which they explained the purpose of the research. Subsequently, the centers contacted families to arrange meetings and explain the purpose of the study. Similarly, some institutions put researchers into contact with families so that they could directly explain the purpose of the study, and an explanatory video was shared on social networks.

The survey was administered online to the parents and caregivers of individuals with ASD following the receipt of proper informed consent. Online surveys were administered through LimeSurvey in Spain and Formsite in Colombia. Both platforms included the same questionnaires. Expert psychologists and similar professionals (educational psychologists and special education teachers) administered the online tests to families. At the beginning of the administration of the tests, the families watched a video explaining the completion of the survey. These professionals helped to resolve any questions that might arise during the survey.

The same Spanish versions of the SCQ-B, RBS-R, PRAS-ASD and PSI-SF were used in both Spain and Colombia. Previously, the cultural equivalence of the items was corroborated. Likewise, the clarity and comprehension of the items was verified with ten Colombian families, and no comprehension difficulties were found. In addition, minor revisions to the lexicon were made in order to cater for regional variations in certain words, such as the Spanish/Colombian use of words for car and door handle, amongst others. Thus, SCQ-B, RBS-R, PRAS-ASD and PSI-SF retro-translations were not necessary for their use with the Colombian sample.

The administration of all questionnaires to families took approximately forty minutes. Questionnaires that were incomplete due to (1) reading comprehension difficulties of the parents (e.g., due to low language skills, low educational level, etc.) and (2) difficulties in connecting to the internet or accessing the web were excluded.

### 2.4. Data Analyses

Statistical analyses were performed using IBM SPSS Statistics v 25.0 for Windows [78]. Means and standard deviations for each sample were obtained from the direct scores provided by participants for each item corresponding to each test. Cronbach alphas were calculated for both countries for all SCQ-B, RBS-R, PRAS-ASD and PSI-SF subscales and overall scales. Next, as a preliminary step to the main analysis, sociodemographic characteristics of the two countries were compared in order to identify potential confounding factors. Chi-square analyses were conducted with the categorical variables of gender, age and comorbidity with IDs. Prior to performing correlations and the comparative analysis of mean values, data distribution was checked in both groups in order to verify fulfillment of the normality assumption required for parametric tests via application of the Kolmogorov–Smirnov test. As this assumption was not satisfied, non-parametric Spearman’s rho coefficients were calculated. Cohen’s criteria [79] were used to evaluate the effect sizes of the correlations. Further, Mann–Whitney *U* tests were performed to determine whether SCQ-B, RBS-R, PRAS-ASD and PSI-SF subscale and overall scale scores differed between Spanish and Colombian subsamples, with differences of *p* < 0.05 being considered significant. Corresponding effect sizes were calculated to determine whether statistically significant differences existed between proportions, with 0.20 ≤ d ≤ 0.50 representing a small effect, 0.51 ≤ d ≤ 0.79 being a moderate effect and d ≥ 0.80 being a large effect [79].

## 3. Results

Characteristics of the 118 participants with ASD from Spain (*n* = 60) and Colombia (*n* = 58) are presented in Table 1. No differences were found regarding sex (χ^2^ = 0.89; *df* = 1; *p* = 0.35). Similarly, no differences emerged regarding age group (4–11 years old and 12–29 years old) (χ^2^ = 0.85; *df* = 1; *p* = 0.36). The outcomes also indicated similarities between both samples in the frequency of ASD diagnosis both with and without intellectual disabilities (IDs), with no statistically significant differences being found between countries as a function of comorbidity with IDs (χ^2^ = 5.59; *df* = 3; *p* = 0.13).

### 3.1. Relationships between Anxiety, RRBs and Parental Stress in Spain and Colombia

The present outcomes revealed a strong positive correlation between RRBs and anxiety in Spanish children with ASD. On the other hand, a moderate association was found between anxiety in Spanish children with ASD and parental stress of caregivers (see Figure 1). Similarly, a moderate association was found between RRBs in Spanish children with ASD and parental stress in their main caregivers (see Appendix A).

With regards to the Colombian sample, a strong positive correlation was also found between RRBs and anxiety in Colombian children with ASD. However, a moderate negative association was observed between anxiety in Colombian children with ASD and parental stress in caregivers (see Figure 2). Similarly, a moderate negative association was found between RRBs in Colombian children with ASD and parental stress in their main caregivers (see Appendix A).

### 3.2. Differences in Anxiety, RRBs and Parental Stress between Spain and Colombia

The present findings indicated that no statistically significant differences exist in the overall SCQ-B scores between Spanish and Colombian individuals with ASD (*U* = 3229.50; *p* = 0.07). Additionally, no differences were found between the two groups in anxiety (*U* = 1421.00; *p* = 0.09) and repetitive behaviors (see Table 2). However, differences did emerge in the parent–child dysfunctional interaction, difficult child and global parenting stress variables (see Table 3), with the Colombian sample with ASD reporting significantly higher mean scores for all subscales and overall PSI-SF. The magnitude of these differences ranged between moderate (DC) and high (P-CDI and GPS).

## 4. Discussion

The present study aimed to analyze the relationships between anxiety, RRBs and parental stress in individuals with ASD in Colombia and Spain, whilst also examining differences in these variables between these two countries. There is a paucity of research comparing anxiety, RRBs and parental stress between Latin American and Western countries (e.g., Spain), highlighting a need to fill this gap in autism research in Spanish-speaking countries [12].

The findings of the present study revealed a strong association between RRBs and anxiety in samples of both Spanish and Colombian individuals with ASD. Furthermore, the variable describing the sameness of behavior was found to produce the greatest correlation with anxiety in both countries. These findings are fully consistent with previous studies [22,29,30,31,32].

Another main finding was that a moderate correlation existed between anxiety and parental stress in the Spanish sample, which coincides with that reported in previous literature [47,48,49]. However, a negative correlation was found between anxiety and parental stress in the Colombian sample of individuals with ASD. This is not a common finding in the available literature [80]. It seems that parenting styles play a key role when it comes to explaining these inconsistencies [81]. In this sense, more punitive, inconsistent or negligent educational styles are associated with negative emotional reactions [81,82,83]. However, the present study did not analyze this variable. Nonetheless, this finding is important to highlight given that it highlights the need to upskill families to equip them with the coping strategies required to deal with the anxiety felt by their children and improve communication. Caregivers of children with ASD face a multitude of challenges and stressors every day. They deal with the inappropriate behaviors and subtle sensory reactions of their children, whilst also taking care of their basic needs, education and treatments. Additionally, they often have to deal with worries about finances and the future of their children. The way in which they deal with these situations, through their parenting style, can help to reduce problematic behaviors and maintain a sense of peace at home [80].

Previous literature has highlighted the relationship between externalizing symptoms, repetitive behavior and parental stress [41,42]. The present study found an association between RRBs and parental stress in the Spanish sample. However, no such relationship was found in the sample of Colombian caregivers with a child with ASD. Again, it is possible that these findings were influenced by the educational style of the participating families [80] who, it is important to note, were not capable of identifying RRB severity or behaviors most associated with moderate or severe anxiety [29,33].

No differences were found in the anxiety and RRBs of children with ASD as a function of the two participating countries in the present research, which is unlike that reported in previous studies [12]. This may be due to the fact that, in the previously mentioned study, the participant groups were not homogenized based on autism severity. However, differences were found in parental stress, with Colombian caregivers reporting higher levels of stress according to the GPI, P-CDI and DC subscales. These findings indicate that Colombian families face more interaction and communication difficulties with their ASD children and are less able to deal with the difficulties this entails. It is also possible that parental stress in Colombian caregivers is influenced by other factors which were not considered in the present study, such as family burden, financial situation, work situation and institutional support.

Despite the progress made by the present study by conducting cross-cultural research in two Hispanic countries, the findings should be considered in light of a series of limitations. Firstly, only a small sample size was recruited. A larger sample size could provide valuable information on whether the country is moderate in parental stress. Secondly, although the sample size was similar to that used in previous studies, differences based on sex or age were not examined due to the small sample size recruited for both countries. This is a very important aspect given that the emotional development of children and their interaction with parents is a process that can change over the years [46]. Thirdly, some variables were not considered, such as the parents’ age, marital status, number of children in the family with and without ASD, families’ economic and employment status and families’ educational level. Fourthly, no relevant instruments were included to analyze the educational styles of parents. Future research should consider all of these variables when examining possible relationships.

These results suggest that it is important to analyze the possible causes of anxiety in autism. Artificial intelligence is a new technological tool that can help to detect possible environmental and biological factors that can influence the emotional development of individuals with ASD [84].

## 5. Conclusions

There is a paucity of cross-cultural studies on autism. It is necessary to promote research on autism in Latin America. This study is an initial step in understanding the levels of anxiety in individuals with autism in Spanish-speaking countries. Anxiety in individuals with ASD may be mediated by psychosocial factors (e.g., parenting style), economic factors and biological factors (e.g., gastrointestinal symptoms).

## Figures and Tables

**Figure 1 brainsci-14-00910-f001:**
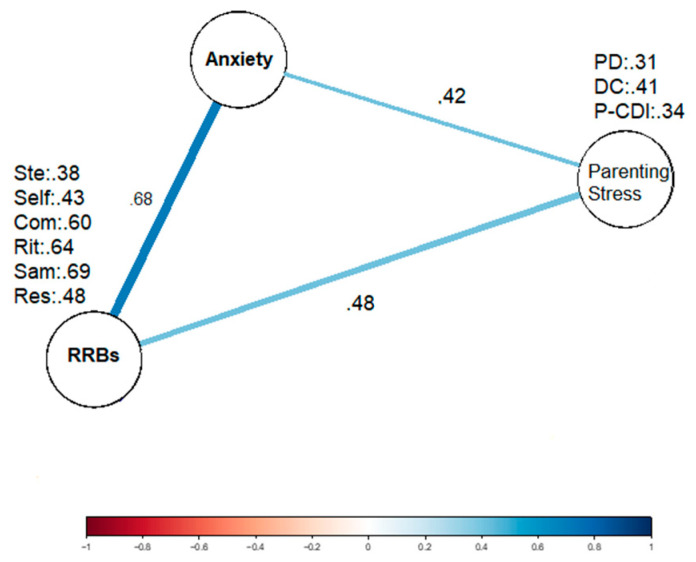
A Gaussian graphical network of the partial correlation matrix of anxiety, repetitive behaviors and parental stress in the Spanish sample of individuals with ASD. Each statistically significant unique partial correlation is illustrated as an edge of color between variables. The width and the color intensity of the edge indicates the strength of the association. Note. Ste = stereotypic; self = self-injurious; Com = compulsive; Rit = ritualistic; Sam = sameness; Res = restrictive behaviors; PD = parental distress; DC = difficult child; P-CDI = parent–child dysfunctional interaction; ** = *p* < 0.01.

**Figure 2 brainsci-14-00910-f002:**
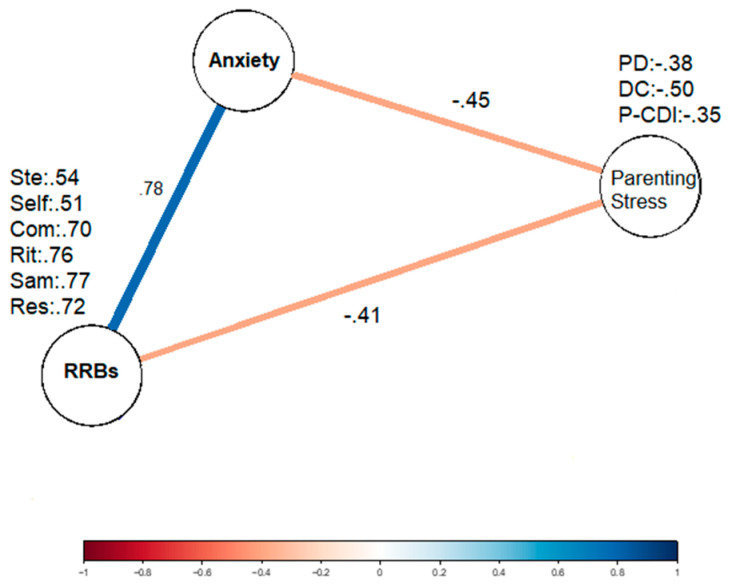
A Gaussian graphical network of the partial correlation matrix of anxiety, repetitive behaviors and parental stress in the Colombian sample of individuals with ASD. Each statistically significant unique partial correlation is illustrated as an edge of color between variables. The width and the color intensity of the edge indicates the strength of the association. Note. Ste = stereotypic; self = self-injurious; Com = compulsive; Rit = ritualistic; Sam = sameness; Res = restrictive behaviors; PD = parental distress; DC = difficult child; P-CDI = parent–child dysfunctional interaction; ** = *p* < 0.01.

**Table 1 brainsci-14-00910-t001:** Sociodemographic and diagnostic characteristics of the sample.

	Spanish Sample		Colombian Sample	
N	60		58	
Age (M/SD)	8.52 (4.41)		10.29 (4.98)	
Sex (male/female)	(46; 76.7%/ 14; 23.3%)		(40; 69%/ 18; 31%)	
**Reported diagnosis**	N	%	N	%
ASD w/o IDs	39	65	46	79.3
ASD w mild IDs	10	16.7	8	13.8
ASD w moderate IDs	7	11.7	4	6.9
ASD w severe IDs	4	6.7	0	0
**Context**	N	%	N	%
Regular class in a regular school	31	51.7	41	70.7
Special class in a regular school	22	36.7	7	12.1
Special school	4	6.7	1	1.7
Other (e.g., residence, day center, etc.)	3	5	9	15.4

Note. w = with; w/o = without; ASD = autism spectrum disorder; ID = intellectual disability; M = mean; SD = standard deviation.

**Table 2 brainsci-14-00910-t002:** Differences in RRBs between Spanish and Colombian individuals with autism.

RBS-R	Spanish Sample	Colombian Sample		
	*M* (*SD*)	*M* (*SD*)	*U*	*p*
Stereotypic	5.38 (4.15)	5.34 (4.26)	1727.50	0.95
Self-injurious	2.55 (3.40)	3.34 (4.47)	1596.50	0.43
Compulsive	5.97 (5.15)	6.19 (6.10)	1678,50	0.74
Ritualistic	5.30 (4.39)	5.33 (5.18)	1652.00	0.63
Sameness	8.90 (6.28)	9.59 (8.55)	1673.50	0.72
Restricted Interests	3.68 (2.77)	4.10 (2.98)	1597.00	0.44
RBS-R Total	31.78 (21.50)	33.88 (27.46)	1727.50	0.95

Note. RBS-R = repetitive behavior scale-revised; *M* = mean; *SD* = standard deviation.

**Table 3 brainsci-14-00910-t003:** Differences in parenting stress according to the country.

PSI-SF	Spanish Sample	Colombian Sample			
	*M* (*SD*)	*M* (*SD*)	*U*	*p*	*d* [95% CI]
Parental Distress	21.70 (9.26)	24.21 (11.22)	1514.50	0.22	-
Parent–Child Dysfunctional Interaction	17.33 (7.54)	31.71 (11.05)	484.50	0.00 ***	−1.52 [1.92; −1.11]
Difficult Child	21.80 (9.27)	26.43 (8.71)	1265.00	0.01 *	−0.51 [0.88; −0.14]
Global Parenting Stress	60.83 (21.78)	82.34 (27.51)	932.50	0.00 ***	−0.86 [1.24; −0.49]

Note. PSI-SF = parenting stress index-short form; *M* = mean; *SD* = standard deviation; * *p* < 0.05; *** *p* < 0.001.

## Data Availability

The raw data supporting the conclusions of this article will be made available by the authors on request. The data are not publicly available due to privacy concerns.

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
