# Peer review of "Relationships between Anxiety, Repetitive Behavior and Parenting Stress: A Comparative Study between Individuals with Autism from Spain and Colombia"

_brainsci, 2024, doi:10.3390/brainsci14090910_

Round 1

Reviewer 1 Report

Comments and Suggestions for Authors

The reason for comparing the Spanish and Colombian cases is not sufficiently explained in the preamble.

The numbers of participants in this study recruited from Spain and Colombia were very small, and it is not sufficient to infer the correlation between children's autistic symptoms and parents' parenting stress in these two countries based on such a small number of participants.

There should be corresponding statistics to check whether the country is a moderator or not.

The authors used parenting styles to interpret the negative association between children’s autism symptoms and parenting stress in Colombian. However, parenting styles were not measured in this study, so this hypothesis is not appropriate.

The research design of this study was too brief to make the findings of the study worthwhile.

Author Response

Comments 1: The reason for comparing the Spanish and Colombian cases is not sufficiently explained in the preamble.

Response 1: The reason why this study has been carried out is due to the scarcity of cross-cultural studies between Spanish-speaking countries. This information was noted at the end of the introduction section (“In conclusion, only a limited number of cross-cultural studies exist on parental anxiety and stress in autistic samples. Most studies compare Western and Eastern cultures, with very few focusing on Latin American populations [49, 50].)

Comments 2: The numbers of participants in this study recruited from Spain and Colombia were very small, and it is not sufficient to infer the correlation between children's autistic symptoms and parents' parenting stress in these two countries based on such a small number of participants. There should be corresponding statistics to check whether the country is a moderator or not.

Response 2: Thank you for pointing this out. We agree with this comment. Future studies with a larger sample size will be able to give us information on whether the country is moderate or not. This limitation is highlighted in the conclusions.

A larger sample size could provide valuable information on whether or not the country is moderate in parental stress.

Furthermore, we point out the limitation of previous studies very similar to ours that also include small samples. The difficulty of accessing the sample of people with autism and the heterogeneity of ASD are well known. Likewise, the analysis of the relationships between these variables with previous studies is justified (page 3 paragraph 2: "Typically, studies include samples of between 20 and 100 caregivers who have a child with ASD”). In addition, this text is added: Likewise, in the analysis of the relationships between between hyperreactivity, RRBs and anxiety previous researchers have a limited sample which can be between 40 and 70 families [22, 24-26]. Similarly, studies with similar sample size have been published for the analysis of relationships between anxiety, RRBs and parental stress in autism [43, 44].

Comments 3: The authors used parenting styles to interpret the negative association between children’s autism symptoms and parenting stress in Colombian. However, parenting styles were not measured in this study, so this hypothesis is not appropriate.

Response 3: Thank you for pointing this out. We agree with this comment. The following information has been removed from the discussion section:

Data from the present study suggest that Colombian caregivers experience lower levels of stress that is not concomitant with the high levels of anxiety felt by their children with ASD. This suggests that the negligent educational style likely prevails.

New references

Shiloh, G., Gal, E., David, A., Kohn, E., Hazan, A., & Stolar, O. (2023). The relations between repetitive behaviors and family accommodation among children with autism: a mixed-methods study. Children, 10(4), 742.

Operto, F. F., Pastorino, G. M. G., Scuoppo, C., Padovano, C., Vivenzio, V., Pistola, I., ... & Coppola, G. (2021). Adaptive behavior, emotional/behavioral problems and parental stress in children with autism spectrum disorder. Frontiers in Neuroscience, 15, 751465.

Fetta, A., Carati, E., Moneti, L., Pignataro, V., Angotti, M., Bardasi, M. C., ... & Parmeggiani, A. (2021). Relationship between sensory alterations and repetitive behaviours in children with autism spectrum disorders: A parents’ questionnaire based study. Brain sciences11(4), 484.

Williams, K. L., Campi, E., & Baranek, G. T. (2021). Associations among sensory hyperresponsiveness, restricted and repetitive behaviors, and anxiety in autism: An integrated systematic review. Research in autism spectrum disorders83, 101763. https://doi.org/10.1016/j.rasd.2021.101763

Black, K. R., Stevenson, R. A., Segers, M., Ncube, B. L., Sun, S. Z., Philipp-Muller, A., ... & Ferber, S. (2017). Linking anxiety and insistence on sameness in autistic children: The role of sensory hypersensitivity. Journal of autism and developmental disorders47, 2459-2470. https://doi.org/10.1007/s10803-017-3161-x

Reviewer 2 Report

Comments and Suggestions for Authors

This paper explores the relationship between anxiety, repetitive behaviors, and parental stress among individuals with autism in Spain and Colombia, comparing differences between the two countries. The study found a strong positive correlation between anxiety and repetitive behaviors in autism populations from both countries. By comparing individuals with autism in Spain and Colombia, the research provides insights from a cross-cultural perspective, aiding in understanding the differences in autism characteristics across different cultural backgrounds. I believe the paper is ready for publication, but certain modifications and improvements must be made before submission.

considering the use of cross-sectional data for model analysis, the study has certain limitations. The introduction and literature review should acknowledge that many studies have utilized longitudinal data, indicating that adolescent development is a dynamic process with heterogeneity among different groups. The authors can refer to and cite the following articles to enhance the introduction and literature review sections, and these references should be used in the discussion section for comparison and analysis of results:

Understanding the Role of ParentChild Relationships in Conscientiousness and Neuroticism Development among Chinese Middle School Students: A Cross-Lagged Model." Behavioral Sciences, 2023, 13, 876. https://doi.org/10.3390/bs13100876

The sample size is small, with only 118 families recruited, which may limit the generalizability of results and statistical power. Further clarification is needed regarding the method of excluding invalid questionnaires, particularly outlining the steps and criteria for exclusion rather than a simple summary.

Gender and age should be considered as potential moderating variables to comprehensively understand differences among different groups.

Other variables such as family economic status, employment status, and educational level that could potentially affect parental stress should be considered.

Instead of just calculating correlations, could regression models be used for further analysis?

Regarding the study conclusions, practical research suggestions should be proposed that are relevant to current applications in artificial intelligence-assisted mental health, which have seen significant advancements.

Author Response

Comments 1: Considering the use of cross-sectional data for model analysis, the study has certain limitations. The introduction and literature review should acknowledge that many studies have utilized longitudinal data, indicating that adolescent development is a dynamic process with heterogeneity among different groups. The authors can refer to and cite the following articles to enhance the introduction and literature review sections, and these references should be used in the discussion section for comparison and analysis of results:

Understanding the Role of Parent‒Child Relationships in Conscientiousness and Neuroticism Development among Chinese Middle School Students: A Cross-Lagged Model." Behavioral Sciences, 2023, 13, 876. https://doi.org/10.3390/bs13100876

Response 1: The contribution is appreciated and is included in both the introduction and discussion sections:

Page 2 paragraph 4: The emotional development of children is a dynamic process over the years and the parent-child relationship is a crucial factor in their mental health [46].

Page 9 paragraph 4: This is a very important aspect given that the emotional development of children and their interaction with parents is a process that can change over the years [46].

 Cao, X., & Liu, X. (2023). Understanding the Role of Parent‒Child Relationships in Conscientiousness and Neuroticism Development among Chinese Middle School Students: A Cross-Lagged Model. Behavioral Sciences13(10), 876. https://doi.org/10.3390/bs13100876

Comments 2: The sample size is small, with only 118 families recruited, which may limit the generalizability of results and statistical power. Further clarification is needed regarding the method of excluding invalid questionnaires, particularly outlining the steps and criteria for exclusion rather than a simple summary.

Response 2: A text with the exclusion criteria for the questionnaires is included in the procedure section (page 5).

The administration of all questionnaires to families took approximately forty minutes. Questionnaires that were incomplete due to: 1) reading comprehension difficulties of the parents (e.g. due to low language skills, low educational level, etc.), and 2) difficulties in the internet connection or web access were excluded.

Comments 3: Gender and age should be considered as potential moderating variables to comprehensively understand differences among different groups. Other variables such as family economic status, employment status, and educational level that could potentially affect parental stress should be considered.

Response 3: We agree with your suggestions. Unfortunately, the small sample size does not allow for moderator analysis. Similarly, this study does not include socio-educational and economic variables of families. These variables are very important and for that reason they have been included as an aspect for future research.

Page 10 paragraph 2: “Thirdly, variables were not considered such as the parents age, marital status, number of children in the family with and without ASD, families’ economic and employment status, and families' educational level”

Comments 4: Instead of just calculating correlations, could regression models be used for further analysis?

 Response 4: We would have liked to include some statistical analysis, however, due to the delivery time this was not possible.

Comments 5:  Regarding the study conclusions, practical research suggestions should be proposed that are relevant to current applications in artificial intelligence-assisted mental health, which have seen significant advancements.

Response 5: Very grateful for your contribution which has been included in the discussion section.

These results suggest that it is important to analyze the possible causes of anxiety in autism. Artificial intelligence is a new technological tool that can help to detect possible environmental and biological factors that can influence the emotional development of individuals with ASD [84].

Climent-Pérez, P., Martínez-González, A. E., & Andreo-Martínez, P. (2024). Contributions of Artificial Intelligence to Analysis of Gut Microbiota in Autism Spectrum Disorder: A Systematic Review. Children11(8), 931. https://doi.org/10.3390/children11080931

Reviewer 3 Report

Comments and Suggestions for Authors

The introduction presents sufficient research data, incl. data from different countries. The authors have transitioned to their own research. As a recommendation for this part, I would say that it is good to add 2-3 sentences as an argument for exactly this type of research and the need to compare those two countries.

2.2.2. Social communication questionnaire, SCQ form B (SCQ-B) The authors must explain who is conducting the research with this tool

2.2.3. Repetitive behavior scale-revised (RBS-R) the same as above and give reference for the Internal consistency of RBS-R in the Spanish sample

2.2.4. Parent-rated anxiety scale (PRAS-ASD) – give more information about the instrument and references. Is this a self-assessment?

2.2.5. Parenting stress index-short form (PSI-SF) – give reference for the Internal consistency

2.3. Procedures and ethics - the procedure for conducting the research is described too vaguely. It is unclear whether and how the researchers met the children, how the parent questionnaires were administered, or who did the IQ tests. This part of the paper needs to be completely rewritten.

It is not clear what the role of this paragraph is: „Expert psychologists and similar professionals (educational psychologists, special education teachers, psychologists) administered the tests to families, supported by their observations and knowledge of individuals with ASD gained from working at the participating institutions serving this population. Psychologists at the center were on hand to help families resolve doubts about diagnosis which were likely to emerge when completing the first part of the survey, whilst an explanatory video highlighted the need for families to consult psychological and psychiatric reports in case of doubt about diagnosis. Researchers organized a training session for all participating centers, at which they de-scribed the purpose of the research study, tests to be used and provided instructions for test administration. and how exactly these activities are included in the whole study.

This text is very disturbing „In addition, minor revisions to the lexicon were made in order to cater for regional variations in certain words, such as the Spanish/Colombian use of words for car and door handle, amongst others.“ Do authors have the right to make revisions in established instruments? Probably not. If so, why were changes made to the questionnaires? What kind of changes have been made?

It is not clear whether the study was approved by an ethics committee.

Results

Nowhere is the age of the participants mentioned, but the results make it clear that the participants are between the ages of 4 and 29. I don't think such a huge difference would be appropriate to include, since the parents have different experiences with raising a child /or adult/ with ASD.

Discussion

I strongly disagree with this statement „In this sense, more punitive, inconsistent or negligent educational styles are associated with negative emotional reactions [76, 77]. A recent study carried out with a sample of neurotypical Colombian children found the most common parenting styles to be authoritarian (38.6%) and neglectful (38%). As a result, it has been observed that a negligent parenting style is associated with lower anxiety [78] Data from the present study suggest that Colombian caregivers experience lower levels of stress that is not concomitant with the high levels of anxiety felt by their children with ASD. This suggests that the negligent educational style likely prevails.. When it comes to children with ASD, this could sound very insulting to parents who endure all the hardships of raising a child with ASD. This part needs to be rewritten.

Author Response

Comments 1:

The introduction presents sufficient research data, incl. data from different countries. The authors have transitioned to their own research. As a recommendation for this part, I would say that it is good to add 2-3 sentences as an argument for exactly this type of research and the need to compare those two countries.

Response 1: We are very grateful for your comment. A transitional sentence has been included on page 3, paragraph 1.

However, there is a paucity of comparative studies on anxiety in children with ASD and parental stress among Spanish-speaking countries.

Comments 2:

2.2.2. Social communication questionnaire, SCQ form B (SCQ-B) – The authors must explain who is conducting the research with this tool

Response 2: In the procedure section it is indicated that the survey is carried out on primary caregivers (“The survey was administered online to the parents and caregivers of individuals with ASD following the receipt of proper informed consent……Expert psychologists and similar professionals (educational psychologists, special education teachers, psychologists) administered the tests to families, supported by their observations and knowledge of individuals with ASD gained from working at the participating institutions serving this population”)

Comments 3: 2.2.3. Repetitive behavior scale-revised (RBS-R) – the same as above and give reference for the Internal consistency of RBS-R in the Spanish simple

Response 3: References have been included

Comments 4: 2.2.4. Parent-rated anxiety scale (PRAS-ASD) – give more information about the instrument and references. Is this a self-assessment?

Response 4: The manuscript stated that it is a scale for caregivers. The reference was also added (“The scale describes a single factor and is administered to caregivers. An α coefficient of .93 has previously been reported [29].)

Comments 5: 2.2.5. Parenting stress index-short form (PSI-SF) – give reference for the Internal consistency

Response 5: Reference has been included

Comments 6: 2.3. Procedures and ethics - the procedure for conducting the research is described too vaguely. It is unclear whether and how the researchers met the children, how the parent questionnaires were administered, or who did the IQ tests. This part of the paper needs to be completely rewritten.

 Response 6: The procedure section has been reorganized.

Comments 7:

It is not clear what the role of this paragraph is: „Expert psychologists and similar professionals (educational psychologists, special education teachers, psychologists) administered the tests to families, supported by their observations and knowledge of individuals with ASD gained from working at the participating institutions serving this population. Psychologists at the center were on hand to help families resolve doubts about diagnosis which were likely to emerge when completing the first part of the survey, whilst an explanatory video highlighted the need for families to consult psychological and psychiatric reports in case of doubt about diagnosis. Researchers organized a training session for all participating centers, at which they de-scribed the purpose of the research study, tests to be used and provided instructions for test administration.“  and how exactly these activities are included in the whole study.

 Response 7: The procedure has been rewritten. We hope it is now written in a clearer way.

Survey administration: In Spain and Colombia, a cover letter containing study information was sent to participating ordinary centers, special education centers, associations of families with children with autism, etc. Researchers organized a training session for all participating centers, at which they described the purpose of the research study, tests to be used and provided instructions for test administration. Many of the tests included in the protocol were known by the professionals at the participating centers. A two-hour session was held to explain the entire protocol and resolve procedural questions to the centers' professionals. Participating centers made appointments with potential participants via an online meeting and individual phone calls at which they explain the purpose of the re-search. Subsequently, the centers contacted families to arrange meetings and explain the purpose of the study. Similarly, some institutions put researchers into contact with families so that they could directly explain the purpose of the study and an explanatory video was shared on social networks.

The survey was administered online to the parents and caregivers of individuals with ASD following the receipt of proper informed consent. The reporting assessment protocol was individually administered through the online survey tool LimeSurvey (LimeSurvey GmbH, Hamburg, Germany) in Spain and Formsite in Colombia. Both platforms included the same comprehensive autism assessment protocol. Expert psychologists and similar professionals (educational psychologists, special education teachers, psychologists) administered the online tests to families. At the beginning of the administration of the tests, the families watched a video explaining the completion of the survey. These professionals helped to resolve any questions that might arise during the survey.

Comments 8: This text is very disturbing „In addition, minor revisions to the lexicon were made in order to cater for regional variations in certain words, such as the Spanish/Colombian use of words for car and door handle, amongst others.“ Do authors have the right to make revisions in established instruments? Probably not. If so, why were changes made to the questionnaires? What kind of changes have been made?

It is not clear whether the study was approved by an ethics committee.

Response 8: Language adaptation is a very important aspect to avoid bias in research and even more so in cross-cultural studies. In our research career, language adaptation has been a previous step to the validation of scales. We have authorization from the main authors of the instruments since we are carrying out the validation of these instruments in Spain (e.g.: PRAS-ASD) and we validate the RBS-R in Spain with authorization from Bodfish (https://aba-elearning.com/guias?id=165). The changes made have been minimal, words that do not affect the grammatical meaning of the items (e.g.: nouns or some verb).

Comments 9: Results

Nowhere is the age of the participants mentioned, but the results make it clear that the participants are between the ages of 4 and 29. I don't think such a huge difference would be appropriate to include, since the parents have different experiences with raising a child /or adult/ with ASD.

Response 9: In Table 1 you can find the mean age and standard deviation of each country. As you can see, there are no age differences between the groups from both countries and both are within the developmental period of childhood (Spanish sample: 8.52 (4.41)/ Colombian sample: 10.29 (4.98)).  

Comments 10: Discussion

I strongly disagree with this statement „In this sense, more punitive, inconsistent or negligent educational styles are associated with negative emotional reactions [76, 77]. A recent study carried out with a sample of neurotypical Colombian children found the most common parenting styles to be authoritarian (38.6%) and neglectful (38%). As a result, it has been observed that a negligent parenting style is associated with lower anxiety [78] Data from the present study suggest that Colombian caregivers experience lower levels of stress that is not concomitant with the high levels of anxiety felt by their children with ASD. This suggests that the negligent educational style likely prevails.“. When it comes to children with ASD, this could sound very insulting to parents who endure all the hardships of raising a child with ASD. This part needs to be rewritten.

Response 10:

This hypothesis has been removed from the discussion. Thanks for your contribution

Round 2

Reviewer 1 Report

Comments and Suggestions for Authors

This study has serious flaws in its research design.

Reviewer 3 Report

Comments and Suggestions for Authors

Thank you for the answers.